# Source-Space Brain Functional Connectivity Features in Electroencephalogram-Based Driver Fatigue Classification

**DOI:** 10.3390/s23052383

**Published:** 2023-02-21

**Authors:** Khanh Ha Nguyen, Matthew Ebbatson, Yvonne Tran, Ashley Craig, Hung Nguyen, Rifai Chai

**Affiliations:** 1School of Science, Computing and Engineering Technologies, Swinburne University of Technology, Melbourne, VIC 3122, Australia; 2School of Engineering, Swinburne University of Technology, Melbourne, VIC 3122, Australia; 3Department of Linguistics, Macquarie University Hearing, Macquarie University, Sydney, NSW 2109, Australia; 4Faculty of Medicine and Health, The University of Sydney, Sydney, NSW 2006, Australia; 5John Walsh Centre for Rehabilitation Research, Kolling Institute, Northern Sydney Local Health District, St Leonards, Sydney, NSW 2065, Australia

**Keywords:** driver fatigue, driving fatigue classification, electroencephalogram, EEG, source space functional connectivity

## Abstract

This study examined the brain source space functional connectivity from the electroencephalogram (EEG) activity of 48 participants during a driving simulation experiment where they drove until fatigue developed. Source-space functional connectivity (FC) analysis is a state-of-the-art method for understanding connections between brain regions that may indicate psychological differences. Multi-band FC in the brain source space was constructed using the phased lag index (PLI) method and used as features to train an SVM classification model to classify driver fatigue and alert conditions. With a subset of critical connections in the beta band, a classification accuracy of 93% was achieved. Additionally, the source-space FC feature extractor demonstrated superiority over other methods, such as PSD and sensor-space FC, in classifying fatigue. The results suggested that source-space FC is a discriminative biomarker for detecting driving fatigue.

## 1. Introduction

Driver fatigue is a major cause of road crashes that can lead to injuries and fatalities [1]. Fatigued driving begins when drivers feel tired and cannot concentrate on normal driving. Their ability to drive significantly decreases, which is commonly reflected through vigilance decline, slower reaction time, and impaired performance, all of which can result in negative outcomes [2]. Therefore, it is essential to have a reliable fatigue measurement as a pragmatic approach to mitigate the risk of fatigued driving [3].

In past decades, countless efforts have been made to measure driver fatigue, including the following: (1) self-reported fatigue questionnaires, (2) video-based measurements that recognize the facial expression, eye blinking frequency, or driving performance indicators such as lane keeping or distance keeping, and (3) physiological measurements that detect fatigue based on physiological indicators such as cardiac rhythms using an electrocardiogram (ECG), eye movements using an electrooculogram (EOG), and brain activities using an electroencephalogram (EEG) [3]. While psychometric questionnaires are direct subjective measures, they are impractical as they require drivers to divert their attention from time to time to an unrelated task besides driving, which is dangerous and counterintuitive. The video-based methods, alternatively, could be affected by ambient conditions and privacy issues, especially with facial recognition. Consequently, physiological measurements seem to be a feasible approach for fatigue monitoring. Past studies have found strong connections between physiological indicators and fatigue; for instance, the high frequency of eye blinking found in EOG can be a sign of sleepiness [4], and changes in heart variability can also be an indicator of tiredness [5]. Among all these physiological measures, EEG is considered to be one of the most predictive and reliable indicators of fatigue as it directly reflects the neurophysiological activities originating in the brain [6].

In general, an EEG-based fatigue classification system consists of a few different components, including (1) data acquisition, which utilizes EEG technology to capture brain signals, and (2) computational intelligence, which applies engineering algorithms such as pre-processing, feature extraction, and classification. For feature extraction, frequency-based analysis is one of the most common EEG analyses, which transforms EEG signals from time domain to frequency domain, such as power spectral density, or PSD [7,8]. The power spectral density analysis converts EEG signals in the time domain to the frequency domain using the Fourier transform. This process can be achieved through Welch’s method [9] or the adaptive multi-taper method [10]. A previous study by Bose et al. showed evidence of the correlation between the PSD and fatigue [7]. However, frequency-based analysis, such as PSD, often estimates brain activity as a whole; thus, it may not provide insightful information on inter-regional interactions in the brain [11].

In recent years, there has been a growing interest in functional connectivity (FC) analysis, which assumes the brain is a functional organization with connections between regions [12,13]. The key advantage of FC analysis is its ability to capture brain activities or neural information transitions from between-regions interactions constructed from EEG signals [14]. In general, FC analysis can be done in (1) sensor space, which directly utilizes EEG signals from scalp-level sensors to compute the FC [15], or in (2) source space, in which the FC is constructed using the neural interactions between source-level regions in the brain [16]. While the sensor-space FC analysis has been used to study cognitive conditions in different applications, such as brain disorder detection [17] and depression detection [18], its reliability has been negatively affected by volume conduction [19,20]. It is mainly due to the effects of instantaneous correlations when electrical activity from a single neuronal source can be picked up by different sensors, meaning that the FC between two signals from two sensors would indicate the activity from the same neuronal source instead of true neuronal interaction [20].

Alternatively, FC can be analyzed in source space to overcome the volume conduction effects [21]. In source-space FC analysis, the neuronal interactions are represented as coupling connections between source-level brain regions [14]. In this regard, EEG signals recorded by the scalp sensors are mapped to electrical activations in source space using a realistic volume conductor head model. These estimated source activations are then parcellated into functional brain regions before a connectivity method is performed to estimate the FC between the regions. Recent studies have shown its potential in representing the dynamics of neuronal processing that underlie different cognition conditions and can be used as discriminative biomarkers in detecting neurological disorders [16,22] and stroke [23]. Due to these promising results in cognitive applications, source-space FC will be used in this study to further explore drivers’ classification of alert and fatigue states.

The novelty of this study lies in the utilization of source-space FC analysis in exploring the dynamic neuronal interactions of drivers during driving. Moreover, the use of source-space FC as a feature extractor for classifying driver alertness vs. fatigue has not previously been explored. The study helps to unmask the potential of source-space FC in detecting driver fatigue compared with other feature extractors. The findings of this study may facilitate the development of a real-time in-vehicle fatigue detection system to improve driving safety in the real world.

## 2. Methods

### 2.1. General Structure

The structure of the proposed EEG-based fatigue classification system is described in Figure 1. Initially, EEG data were collected from a simulated driving experiment before going through pre-processing steps to remove EEG artifacts before being filtered in five different frequency bands: delta (0.5–4 Hz), theta (4–7 Hz), alpha (8–12 Hz), beta (13–30 Hz), and gamma (32–45 Hz). Consequently, the feature extraction component transformed the signals into useful features using source-space FC analysis. These features were used as inputs for training machine learning models for classification between fatigue and alert state.

### 2.2. EEG Recordings and Pre-Processing

The EEG data used in this study were obtained from a previous study [24]. The dataset contains EEG signals from 48 healthy participants in a driving experiment using the Divided Attention Steering Simulator (DASS) from Stowood Scientific Instruments. In the experiment, participants were asked to perform a monotonous task that involved driving in the center of the road for a maximum of 2 h or until fatigue appeared. To account for the impact of circadian rhythms, all participants underwent testing during either 9:00 a.m. to 12:00 p.m. or 2:00 p.m. to 5:00 p.m. [24]. These two time slots were chosen as they have been demonstrated to have comparable effects on circadian rhythms [25]. Fatigue was confirmed based on either (1) fatigue indications such as nodding, yawning, prolonged eye closure, and eye twitching or (2) when the participant deviated off the center of the road for more than 15 s. A drowsiness questionnaire using the Stanford Sleepiness Scale was employed before and after the task as a validation for the development of fatigue in participants [26].

The EEG data were collected using the Active-Two, a 32-channel EEG system from Biosemi. Electrodes were placed based on the international 10-20 standard [27]. These positions are: *FP1*, *AF3*, *F7*, *F3*, *FC1*, *FC5*, *T7*, *C3*, *CP1*, *CP5*, *P7*, *P3*, *PZ*, *PO3*, *O1*, *OZ*, *O2*, *PO4*, *P4*, *P8*, *CP6*, *CP2*, *C4*, *T8*, *FC6*, *FC2*, *F4*, *F8*, *AF4*, *FP2*, *FZ,* and *CZ*. The EEG signals were then downsampled from 2048 Hz to 256 Hz.

For each participant, two sets of 20 s EEG data from the beginning and end of the driving task were selected to represent the alert and fatigue states, respectively. Subsequently, the EEG signals were pre-processed using second-order blind identification (SOBI) and canonical correlation analysis (CCA) to remove artifacts related to eye movements, muscle activities, and cardiac signals. Clean EEG data were then decomposed into delta (0.5–4 Hz), theta (4–7 Hz), alpha (8–12 Hz), beta (13–30 Hz), and gamma (32–45 Hz) frequency bands for further analyses.

### 2.3. Feature Extraction: Source-Space FC Analysis

The analytic stream of the source-space FC can be summarized in the following steps: (1) reconstruction of the source activation by projecting clean EEG signals into a realistic head model; (2) parcellation of estimated source activities into regions of interest (ROIs) in main lobes; (3) estimation of FC between brain ROIs. In this study, FC analysis was computed using the open-source library MNE-Python [28].

#### 2.3.1. Source Localization

Source localization is a process used to locate the origin of electrical activity in the brain using EEG recordings taken from the scalp. It consists of two steps: (1) creating a leadfield matrix, which shows the relationship between electrical activity in the brain and the signals measured at the scalp electrodes, using a head model to simulate the electrical sources (*forward problem*); and (2) using the leadfield matrix to determine which brain regions produced the EEG signals observed at the electrodes (*inverse problem*).

Calculating the “forward solution” requires constructing a conductivity model that simulates an anatomical head’s geometries and electrical conductivities. This study used the “fsaverage” conductor head model, which is reconstructed based on the “Buckner40” dataset. The head model was provided by FreeSurfer using the boundary element method (BEM) [29], which is a more realistic head model compared with the simple spherical head model [30]. See this study [30] for a comparison between different methods to construct conductor head models. The BEM model consists of three layers: inner skull, outer skull, and outer skin, which define compartments of tissues in the head (Figure 2). Since the BEM model is constructed from the magnetic resonance image (MRI) data, which were collected by an MRI device, it is necessary to align it with the EEG electrodes so that they are in a common coordinate system; the aligning process is also known as coregistration. For this, EEG electrode positions were retrieved from the standard 10–20 montage and projected onto the head model based on fiducial markers, which are anatomical locations as the reference; the BEM model is visualized in Figure 3.

After developing the head model with projected EEG sensors, the next step is creating a source space, a predefined set of current dipoles distributed over the cerebral cortex. A current dipole is a source-point model approximating the flow of electrical current generated by neurons in a small cortical region (source vertex) [31]. In each source vertex, three dipoles present three orthogonal orientations, i.e., XYZ-coordinates. The forward solution is then solved by using a forward operator to calculate the electric potential at each sensor from a given electrical current dipole, resulting in a gain or lead field matrix. For this study, a surface-based source space with the ‘*oct5*’ resolution was chosen, which contains 10,242 vertices per hemisphere or 20,484 vertices for both hemispheres (source space within the BEM head model, which is shown in Figure 4). Finally, there will be 61,452 dipoles fed through the forward operator, resulting in a leadfield matrix with the shape of 32 sensors × 61,452 dipoles.

The lead field matrix found is then used to form an inverse operator to estimate the brain source activity (inverse problem). Various methods can be used for the inverse problem; some common methods are MNE (Minimum Estimate of the standard) [28], sLORETA (standardized Low Resolution Electromagnetic Tomography) [32], or dSPM (dynamic Statistical Parametric Mapping) [33]. In this study, sLORETA was selected for the inverse problem as it shows advantages over other methods [34]. sLORETA is based on standardizing the current density estimated for source localization and has been proven to provide zero localization error [32]. Subsequently, the inverse solution was solved by feeding the EEG time series into the inverse operator. This resulted in a time series of activations at source vertices, that is, source estimates. The output of the inverse solution is a matrix with a shape of $N_{vertices}\times N_{time-series}$, where $N_{vertices}$ is the number of vertices and $N_{time-series}$ is the number of time points in the time series.

#### 2.3.2. Parcellation of Source Vertices to ROIs

Analysis at the vertex level is complicated and exhausting; therefore, combining neighboring vertices close to each other is necessary to form regions of interest. Using brain atlases, these clusters can be parcelled into ROIs, which are functionally and structurally distinct brain regions, such as the brain lobes.

In this study, FreeSurfer’s ‘*aparc*’ cortical atlas parcellation with 34 ROIs per hemisphere was used [35,36]. The representative activation of each ROI was calculated by averaging the source estimates across all the vertices within that ROI. To avoid signal cancellation, a *sign-flip* will be applied to the source estimates at vertices whose orientation is more than 180° different from the dominant orientation (the vector orientation in which the signal magnitude is largest). The brain ROIs are shown in Figure 5.

The source estimates after parcellation will be a time series of the activation at brain ROIs, which is a matrix with the shape of $N_{ROIs}\times N_{time-series}$, where $N_{ROIs}$ is the number of ROIs, which is 68 (34 × 2 hemispheres) in this study. This matrix will be used to estimate the FC between ROIs.

#### 2.3.3. FC Estimation

FC is generally a measure of neural interactions between different brain regions. While neural interactions can be presented as amplitude and phase dynamics [14], this study will focus only on the phase-based correlations as phase-to-phase synchrony reflects the mechanism of neuronal communication [37]. We employed the sliding window technique to calculate the FC matrices, which allowed the capture of the dynamic FC [38]. This required EEG signals to be split into a set of temporal windows. In this study, window length was determined by following the smallest window length rule suggested in the study [39], in which the window length is greater than number of cycles/central frequency, which was set to 5 by default in the MNE-Python library [28].

Subsequently, the Phase Lag Index (PLI) was chosen as the method to estimate the connectivity based on the cross-power spectral density (CPSD) between source activation time series:$$PLI=[E[sign\left(Im\left(S_{xy}\right)\right)]]$$
where E denotes the average over data epochs and $S_{xy}$ is the CPSD between time series x(t) and y(t), where $S_{xy}$ is defined as the Fourier transform of the cross-correlation function $R_{xy}\left(\tau \right)$:$$S_{xy}\left(f\right)=F\left(R_{xy}\left(\tau \right)\right),\;\mathrm{with}\;R_{xy}\left(\tau \right)=\underset{T\to \infty }{\lim}\frac{1}{T}\int_{0}^{T}x\left(t\right)y\left(t-\tau \right)dt$$

In the PLI method, the synchronization between signals is presented via the frequency of leads (ahead) or lags (behind) between them, in which a PLI value of 0 means x(t) leads or lags y(t) equally often; a PLI value greater than 0 means there is an unbalanced likelihood of x(t) being leading or lagging y(t), while 1 means x(t) always leads or lags y(t).

PLI was applied to estimate the correlation between all pairs of source estimate time series, resulting in a square matrix of size $N_{ROIs}\times N_{ROIs}\;\left(68\times 68\right)$, in which the value at the *xth* row and *yth* column is the connectivity value between the *xth* ROI and the *xth* ROI. Values at the matrix diagonal will be all 0, as they are self-correlations, and as the PLI connectivity is bidirectional, x-y connection and y-x connection are identical; thus, the connectivity values in the lower-triangle part and upper-triangle part will be the same. Therefore, only the lower-triangle parts of the PLI connectivity matrices were used for the classification between fatigue and alert state.

#### 2.3.4. Other Feature Extractors

For comparison purposes, power spectral density (PSD) and sensor-space FC were used in this study. With power spectral density, Welch’s method was applied to estimate the power spectrum of the EEG epochs:$${\hat{p}}_{W}\left(f\right)=\frac{1}{N}\sum_{n=0}^{N-1}{\hat{p}}_{n}\left(f\right)$$
where ${\hat{p}}_{W}\left(f\right)$ is the Welch’s PSD estimation, N is the number of epochs, ${\hat{p}}_{n}\left(f\right)$ is the periodogram estimate of the *nth* epoch. Then the relative PSD of each frequency band was extracted by normalizing the PSD of one band to the whole frequency band.
$${\hat{p}}_{relative}\left(f\right)=\frac{\sum_{f=f_{1}}^{f=f_{2}}{\hat{p}}_{W}\left(f\right)}{N\sum_{f=f_{H}}^{f=f_{H}}{\hat{p}}_{W}\left(f\right)}$$
where $\;\left[f_{L},f_{H}\right]=\left[0.5,\;45\right]$ and $\left[f_{1},f_{2}\right]$ is the low bound and high bound frequency of the selected frequency sub-band, including delta (0.5–4 Hz), theta (4–7 Hz), alpha (8–12 Hz), beta (13–30 Hz), and gamma (30–45 Hz). This yielded 32 features corresponding to 32 EEG sensor signals.

With sensor-space FC, PLI was used to estimate the connectivity between EEG sensor signals. This resulted in connectivity matrices of size 32 channels × 32 channels for five analysis frequency bands, which contained PLI values of pair-wise connectivity among all EEG channels. Similar to source-space FC, only the lower-triangle parts of the connectivity matrices were used as features for fatigue/alert classification.

### 2.4. Classification Alert vs. Fatigue

#### 2.4.1. Feature Selection

Connectivity matrices were used as features for fatigue/alert classification. With the source-space connectivity matrices, there were  68×(68−1)/2=2278 pair-wise ROI-level connections per frequency band, per cognitive condition, and per subject. Considering the high number of features, it is necessary to implement a feature selection to remove redundant features and avoid possible overfitting of the model [40]. Here, we employed recursive feature elimination (RFE), which uses a machine learning algorithm to rank the features according to their contribution to the accuracy of the cognitive state classification. The process iteratively removed the least important features from the current set of features until a certain number of valid features was left [41]; these remaining features were considered critical connections.

#### 2.4.2. Classification Algorithm

The classification was performed with a Support Vector Machine (SVM) classifier. The SVM classifier operates by mapping samples to points in a multi-dimensional plane and finding a gap or a hyper-plane to divide the sample plane into multiple parts representing different categories; each sample point is then assigned to one of the categories [42]. Initially, SVM was designed for linear classification; however, it can also be used for non-linear classification by implementing non-linear kernels. Basically, non-linear kernels such as polynomial, sigmoid, and Gaussian RBF kernels will implicitly transform the inputs into higher-dimensional spaces so that the SVM algorithm can find the optimal hyper-plane in a more complex space [43]. Depending on the characteristics of features, different kernels can result in different classification performances; hence, it is important to choose a suitable kernel to improve the accuracy of the classification algorithm. In order to achieve the best results, some techniques for adjusting hyper-parameters can be utilized [44]. In this study, a Grid Search technique was used for hyper-parameter tuning, which searches over a parameter grid to find the optimal set of parameters resulting in the best results [45].

To reduce the possible effect of the training set variability, classification was performed using 1000 repetitions of two-fold cross-validation with all data [46]. The classification results were then calculated as an average of all repetitions and will be presented as performance metrics such as sensitivity, specificity, and accuracy:$$Sensitivity=\frac{TP}{TP+FN}$$
$$Specificity=\frac{TN}{TN+FP}$$
$$Accuracy=\frac{TP+TN}{TP+TN+FP+FN}$$
where TP is True fatigue Positive, which is the number of fatigue samples that were truly predicted as fatigue; TN is True fatigue Negative, which is the number of alert samples that were correctly predicted as alert; FP is False fatigue Positive, which is the number of samples that were predicted as fatigue but belonged to alert; and FN is False fatigue Negative, which is the number of fatigue samples that were misclassified as alert. As a result, sensitivity can also be inferred as True Positive Rate, which indicates the probability that a model will correctly predict fatigue conditions, and specificity can be interpreted as the probability that the model will correctly predict alert conditions.

The dataset includes EEG signals from 48 participants, with each participant having one 20 s EEG signal for the alert state and another 20 s EEG signal for the fatigue state. Moreover, considering the small sample size, a permutation test was employed to study the significance of the classification results, in which classification was applied 1000 times with random permutations of class labels to estimate the distribution of classification accuracy. A *p*-value is then computed as the ratio of the number of classifications trained on random labels that have better accuracy than the classification trained on the original labels to the total number of permutations [47]. As a result, the *p*-value denotes the probability that the classification performs better with randomly shuffled labels than with actual labels. In this study, a significance level of 0.05 was selected.

## 3. Results

### 3.1. Feature Selection

For the RFE, different numbers of features in terms of percentage, starting from 1% to 99%, were examined to choose the optimal number of features for the classification algorithm. As the number of source-space FC features in this study is 2278, each percentage represents roughly 22 connectivity features.

Figure 6 shows the classification accuracy with different numbers of features in five EEG bands. The highest accuracy was archived in the beta band, in which the accuracy curve starts at nearly 80% with 1% features selected and increases significantly to more than 90% at 5% features before reaching the highest peak at 93% with 13% of total features selected. High accuracy above 90% was maintained until 25% of the features began dropping quickly to below 70% with half of the features. As a result, the optimal number of features was 13%.

### 3.2. Effect of Fatigue on Critical Connections

Selected FC features were further examined. Initially, differences in connection strength were calculated by subtracting connectivity values between alert and fatigue conditions. As a result, positive values (increased-strength connections) denoted higher connectivity strength in the fatigue state compared with the alert state, and negative values (decreased-strength connections) meant the opposite. As shown in Figure 7, increasing trends in connectivity strength were observed in delta (162/134), theta (156/140), and alpha (164/132), while the number of increased-strength connections was significantly lower than that of decreased-strength connections in beta (132 increased/164 decreased) and gamma band (113 increased/183 decreased).

With the distribution of critical connections, the majority of the connectivity features were connections located in the frontal lobe, which accounted for 40.5% in delta, 35.5% in theta, 35.1% in alpha, 32.1% in beta, and 37.5% in gamma (Figure 8, Figure 9, Figure 10, Figure 11 and Figure 12). Parietal-related connections were the second largest with 27.3% on average, especially in the beta band, which accounted for 33.4% of the critical connections.

When fatigue occurred, some connections were strengthened, and some connections were weakened. With increased-strength connections, frontal-related connections took the largest share, with 43.8% in delta, 30.8% in theta, 36% in alpha, 33.3% in beta, and 49.6% in gamma of the total increased-strength connections. Furthermore, intra-frontal connections (connections within the frontal region) outnumbered the intra-connections of other regions, with 50% in delta, 51% in theta, 43% in alpha, 56% in beta, and 76% in gamma. Alternatively, decreased-strength connections were found to spread across all brain regions. Especially, a significant decrease of parietal-related connections was observed in the beta band, which accounted for 42.1% of total decreased-strength connections (Figure 11). Interestingly, inter-regional connections (e.g., frontal-parietal or frontal-temporal) accounted for a major part of the total decreased-strength connections, with 68.7% in delta, 77.8% in theta, 72.1% in alpha, 77.4% in beta, and 67.7% in gamma.

### 3.3. Classification Results

Subsequently, classification was performed with 13%, or 296 connectivity features, selected from the original 2278 connectivity features. The SVM model was trained 1000 times with a two-fold cross-validation approach. Classification results were reported as average accuracy, average sensitivity, and average specificity for 1000 repetitions. Table 1 presents the classification results of the SVM model in five frequency bands. Classification using connectivity features in the beta band archived the highest accuracy of 93% (*p* < 0.001). Moreover, sensitivity and specificity of classification in the beta band features were also the highest among all frequency bands, at 94% and 93%, respectively. Classification in the theta band features the second-highest accuracy with 90% sensitivity, 91% specificity, and 90% accuracy (*p* < 0.001). Both delta and gamma frequency bands scored an accuracy of 88% (*p* < 0.001), while alpha band connectivity features yielded the lowest classification results with 77% (*p* < 0.001) accuracy. With all EEG bands, the *p*-value obtained from the permutation tests was smaller than the significance level of 0.05, indicating a low probability that the achieved classification accuracy happened by chance.

Next, source-space FC features were compared with features from other feature extractors, such as the PSD extractor and the sensor-space FC extractor, in distinguishing between alert and fatigue. With the PSD extractor, Welch’s PSD estimation was used for the PSD features to extract the power of EEG signals in different frequency bands, following the workflow suggested by the study [2]. With sensor-space FC, the PLI method was employed with EEG electrode signals for the sensor-space FC to produce the sensor-level FC. All the features were then used for training a SVM classification algorithm with a cross-validation grid search to find the best model parameters. The classification was iterated 1000 times, and the average accuracy of different feature extractors is illustrated in Figure 13. A permutation test was also employed to assess the significance of the classification accuracy.

With PSD feature extractors, the highest classification accuracy was achieved in the alpha band at 73% (*p* < 0.001), while classification with sensor-space FC features obtained the highest accuracy of 85% in the beta band (*p* < 0.001). The best performance of source-space FC features was in the beta band with 93% accuracy (*p* < 0.001). Furthermore, among all frequency bands, classification using the source-space FC feature extractor achieved the highest accuracy, ranging from 77% to 93%, while the accuracy range of the sensor-space FC extractor was 71% to 81%, and it was only 60% to 73% with the PSD extractor. Overall, the best feature extractor for classification between alert/fatigue was source-space FC in the beta frequency band.

## 4. Discussion

In this study, we explored driver fatigue classification using source-space FC features. This cutting-edge FC analysis allowed the computing of neuronal connections between brain regions, which may help reduce the effect of volume conduction in scalp-level FC methods. Using a realistic head model, source localization was performed to construct the brain electrical activation map, or source estimates, which were later used for an inverse solution to estimate the FC between regions in the brain. The RFE inverse solution was employed as a feature selection approach to improve classification performance and avoid possible overfitting problems by reducing unnecessary connections before classification. Subsequently, the hyperparameter tuning grid search was utilized to find the best parameter to initialize the SVM classifier. The classification was performed using 1000 repetitions of two-fold cross-validation on the whole dataset. This helped to avoid possible training set variability. Moreover, considering the small sample size, a permutation test was performed to assess the significance of the classification accuracy. Classification results suggested that source-space FC in the beta frequency band were discriminative features for classification between alert and fatigue.

The discriminative features for fatigue detection corresponded to the source-space FC in the beta band. This result aligned with the previous study by Wang et al., in which differences between alert and fatigue states correlated with changes in brain activities in the beta band [11]. Moreover, the comparison between different feature extractors suggested that classification using source-space FC achieved the highest accuracy, followed by sensor-space FC and PSD, respectively. This suggested that FC-family analysis could be better at providing representative features reflecting the difference between cognitive states, such as alert vs. fatigue, compared with conventional frequency analysis. Previous studies have adopted the FC analysis in classifying cognitive states; for example, sensor-space FC features were used to detect driving distraction [48] or mental stress [49]. Furthermore, considering methods in the FC family, source-space FC provided better connectivity features for fatigue detection than sensor-space FC. This was due to the superiority of source-space FC over sensor-space FC with volume conduction [14]. With such results, source-space FC could properly capture the differences in how signals transmit within brain regions and could be suitable for developing fatigue detection applications.

Differences in connection strength were found between alert and fatigue conditions. With the presence of fatigue, increasing trends in connectivity strength were found in delta, theta, and alpha, while there were decreases in connectivity strength in beta and gamma. As connectivity strength represents the degrees of synchronization between sources, more increased connections denote higher synchronization and, subsequently, higher brain activity [50]. As a result, this finding suggested that when fatigue occurred, there would be an increase in brain activity in low-frequency components (delta, theta, and alpha), while lower brain activity could be observed in the mid- and high-frequency components (beta, gamma). Those results are consistent with those of other studies and suggest that an increase in brain activity in the alpha band was associated with increased fatigue levels [51]. Moreover, a decline in brain activity in beta was reported to be a sign of fatigue [52,53].

The distribution of connections also provided a clearer picture of the differences in the cognitive mechanism between alert and fatigue states. A large share of critical connections were reported to be related to the frontal regions. Frontal-related connections increased in strength when fatigue occurred, especially connections within the frontal region, or intra-frontal connections. Such results reflected a concentrated distribution of increased strength connections in the frontal area, which is generally responsible for attention sustainability [11,54,55]. Alternatively, decreases in connection strength were found across all brain regions. Typically, inter-regional connections such as frontal-parietal and frontal-temporal accounted for a major part of the decreased strength of connections. Such findings are in line with those obtained by other studies indicating a decline in brain activity on a global scale associated with the presence of fatigue [11,56].

## 5. Conclusions

In this study, the two-state classification (alert vs. fatigue) was employed with EEG recordings from 48 participants in a simulated driving experiment. Source space-FC was used as a feature extractor. The highest classification result of 93% was achieved with the SVM model in beta-band critical connections with RFE as the feature selection. The distribution of critical connections suggested that lower synchronization of brain signals happened in the low EEG bands (delta, theta, and alpha) and higher synchronization happened in the high EEG bands (beta, gamma), which were associated with the presence of fatigue. Furthermore, when drivers felt tired, there was an increase in the critical connections related to the frontal lobe, which played a crucial part in sustaining attention. Overall, the findings of this paper revealed critical source-space FC features for detecting driver fatigue. This could pave the way for the development of a realistic driving fatigue detection system using portable EEG.

In this research, EEG signals for alert and fatigue conditions were not continuous. The alert signals were collected at the start of the experiment when drivers were alert, while fatigue signals were taken at the end of the experiment when drivers were fatigued. The next step for the study is to explore the transition from alertness to fatigue, as this would provide deeper insight into the development of fatigue during driving. Furthermore, combining FC features with other brain network features such as graph theoretical analysis for driver fatigue detection may be worth investigating as it would capture neuronal interactions between brain regions locally and globally.

## Figures and Tables

**Figure 1 sensors-23-02383-f001:**
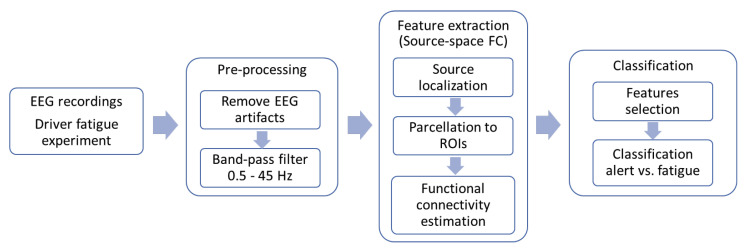
Block diagram of the EEG–based driver fatigue classification using source-space FC features.

**Figure 2 sensors-23-02383-f002:**
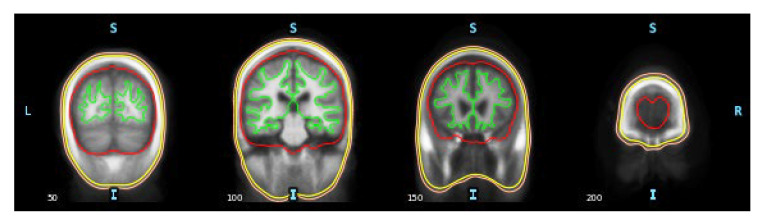
BEM head model with the outer skin layer in orange color, the outer skull layer in yellow color, the inner skull layer in red color, and the brain surface in green color. Orientation is shown as L (Left), R (Right), S (Superior), and I (Inferior).

**Figure 3 sensors-23-02383-f003:**
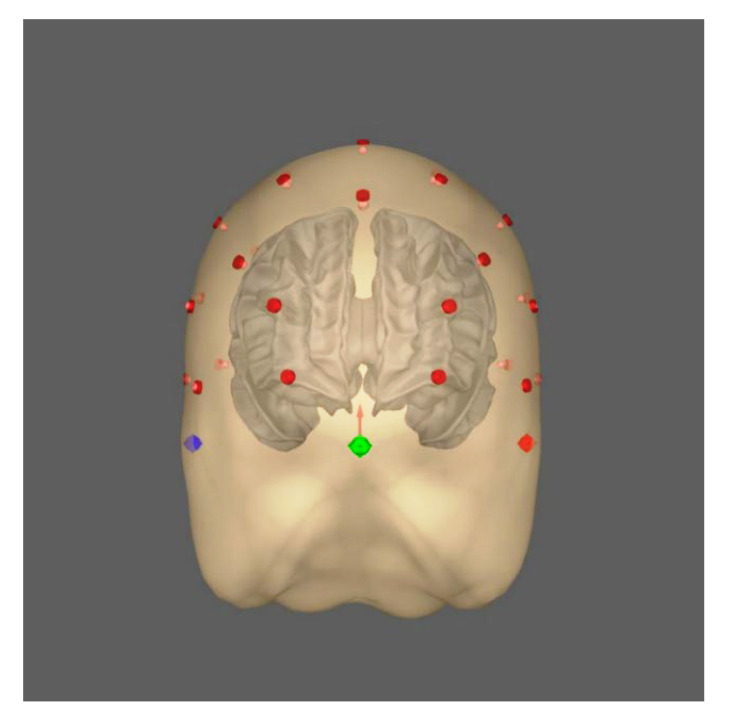
Coregistration of EEG sensor locations in the head model. Red points are EEG electrodes; blue and green points are the fiducial markers.

**Figure 4 sensors-23-02383-f004:**
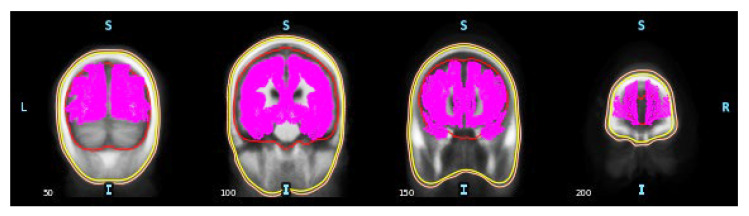
Source space visualization. The outer skin layer is in orange color, the outer skull layer is in yellow color, the inner skull layer is in red color, the vertices are in pink color. Orientation is shown as L (Left), R (Right), S (Superior), and I (Inferior).

**Figure 5 sensors-23-02383-f005:**
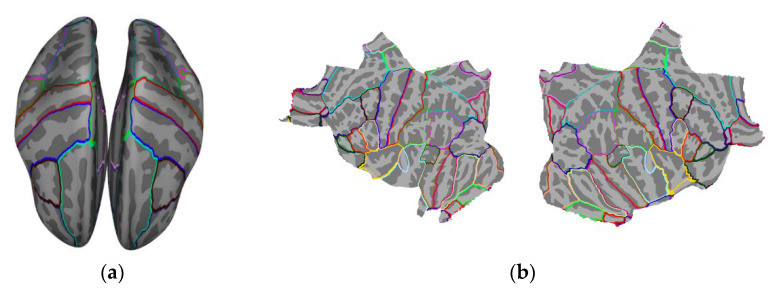
Brain ROIs visualization in (**a**) inflated brain view and (**b**) flat view.

**Figure 6 sensors-23-02383-f006:**
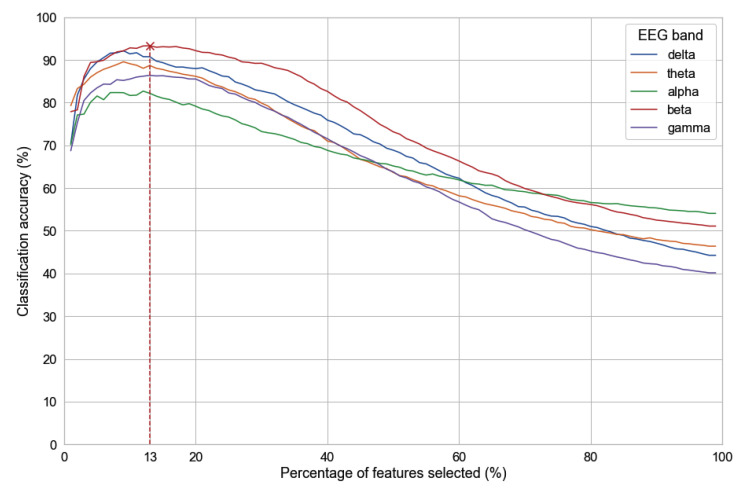
Finding the optimal number of features to plot.

**Figure 7 sensors-23-02383-f007:**
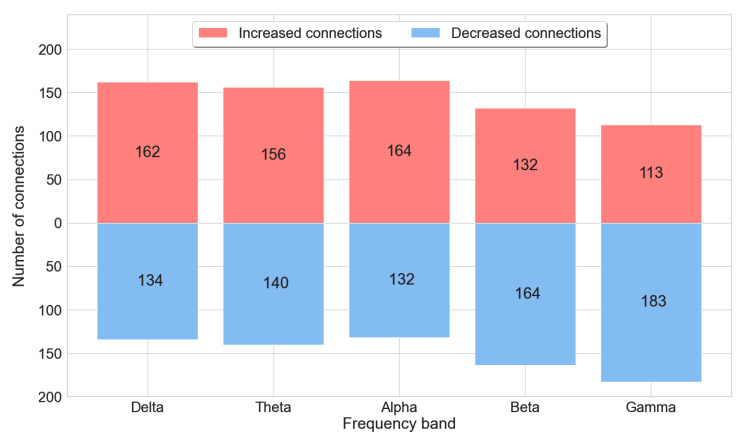
Number of increased-strength connections and decreased-strength connections between alert and fatigue states in five EEG bands. Increased-strength connections denote a higher connection strength in the fatigue state than in the alert state. Decreased-strength connections denote a lower connection strength in the fatigue state than in the alert state.

**Figure 8 sensors-23-02383-f008:**
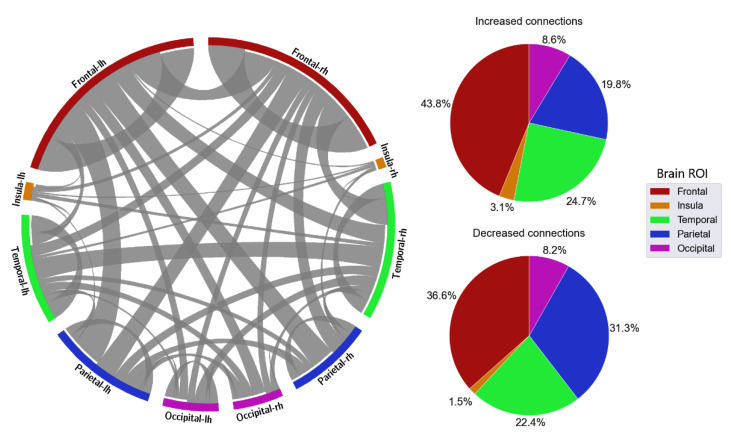
FC and its distribution in different brain ROIs in the delta band. *lh* denotes the left hemisphere, and *rh* denotes the right hemisphere.

**Figure 9 sensors-23-02383-f009:**
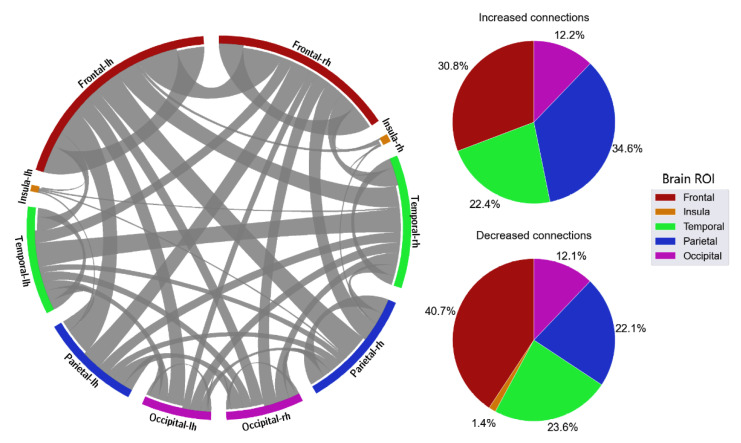
FC and its distribution in different brain ROIs in the theta band. *lh* denotes the left hemisphere, and *rh* denotes the right hemisphere.

**Figure 10 sensors-23-02383-f010:**
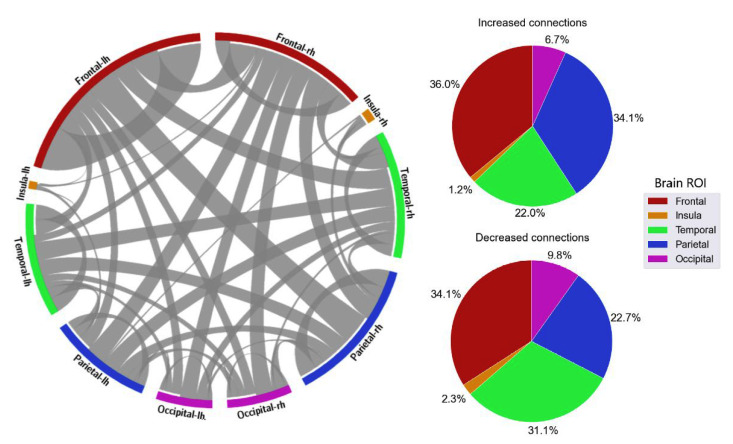
FC and its distribution in different brain ROIs in the alpha band. *lh* denotes the left hemisphere, and *rh* denotes the right hemisphere.

**Figure 11 sensors-23-02383-f011:**
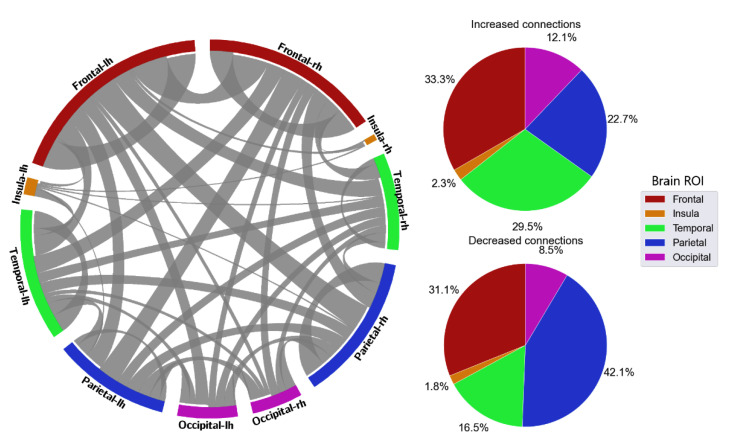
FC and its distribution in different brain ROIs in the beta band. *lh* denotes the left hemisphere, and *rh* denotes the right hemisphere.

**Figure 12 sensors-23-02383-f012:**
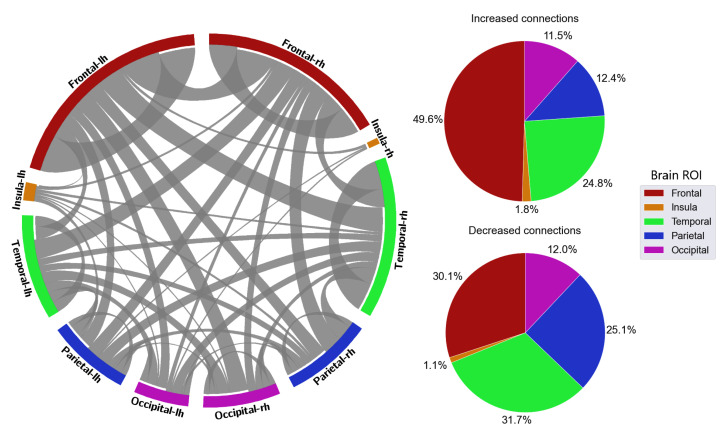
FC and its distribution in different brain ROIs in the gamma band. *lh* denotes the left hemisphere, and *rh* denotes the right hemisphere.

**Figure 13 sensors-23-02383-f013:**
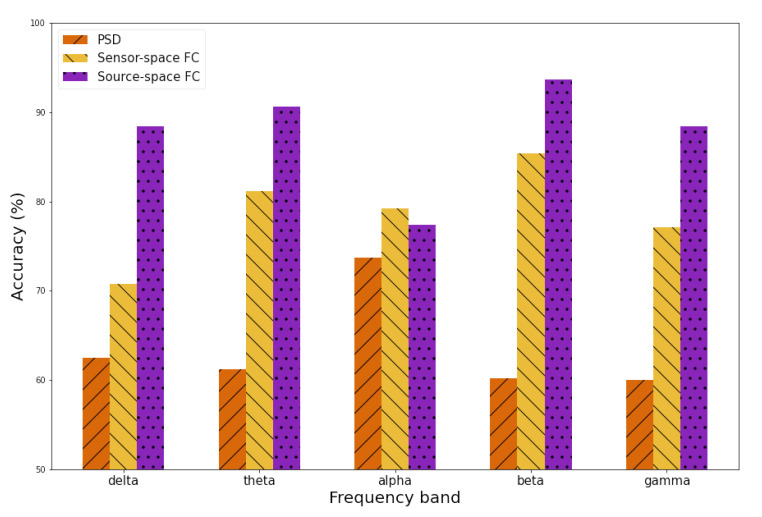
Comparison of classification performances with features among PSD, sensor-space FC, and source-space FC.

**Table 1 sensors-23-02383-t001:** Alert/Fatigue classification results with source-space FC features in 5 frequency bands. Feature selection (RFE) was applied to select 296 connectivity features (13%) from the original 2278 connectivity features. Classification results are reported in Sens (Sensitivity), Spec (Specificity), Acc (Accuracy), and *p*-value of the permutation test.

EEG Band	Average Results of 1000 Classification Iterations			
	Sens	Spec	Acc	*p*-Value
Delta (0.5–4 Hz)	88%	88%	88%	<0.001
Theta (4–7 Hz)	90%	91%	90%	<0.001
Alpha (8–12 Hz)	88%	67%	77%	<0.001
Beta (13–30 Hz)	**94%**	**93%**	**93%**	**<0.001**
Gamma (32–45 Hz)	83%	92%	88%	<0.001

## Data Availability

For third-party data, restrictions apply to the availability of these data. Data were obtained from the University of Technology Sydney and are available from A. Craig et al. or at https://www.sciencedirect.com/science/article/pii/S0301051105001225 (accessed on 25 September 2022) with the permission of the University of Technology Sydney [24].

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
