# Peer review of "Source-Space Brain Functional Connectivity Features in Electroencephalogram-Based Driver Fatigue Classification"

_sensors, 2023, doi:10.3390/s23052383_

Round 1

Reviewer 1 Report

The topic is interesting, and the research paper has wider theoretical and practical applications. The authors have put their best efforts into executing this paper. However, I have the following reservations and suggestions for the sake of improvement of the undertaken study:

1) The logical sequence of the abstract should be as 1) objectives, 2) methodology, 3) Findings, 4) conclusion, and 5) implications. Thus, the authors should also rewrite the abstract in this sequence. 

2) The authors did not establish the study's motivation, significance, objectives, and novelty. The authors are suggested to improve these important factors in the "Introduction" section. 

3) The literature should be presented in separate heading, it should be presented in an audit form and should be linked with the objectives of the current paper. 

4) The conclusion needs further elaboration, and it should always be one step ahead of the findings. 

5) The practical, theoretical and societal implications should be discussed after the conclusion and in light of the conclusion and discussions. 

6) The limitations and areas of future research studies should be discussed in the end of the paper.

7) Minor spelling and grammatical mistakes should be improved.

Author Response

Please refer to the attached file for the response to Reviewer 1.

Reviewer 2 Report

1. I understood that the study used data obtained in a previous study, but did it not receive research ethics approval? If so, I think you should clearly state the approving agency and approval number.

2. I would like to ask about sample size. What was the rationale for this sample size?

3.Is the time of day (i.e., morning/afternoon, not the time of measurement) the same for all subjects when the EEG was measured? If not, is there any difference between the morning and afternoon measurements? Has that been verified? Or are there any previous studies that mention it?

Author Response

Please refer to the attached file for the response to Reviewer 2.

Reviewer 3 Report

This paper analyzed source space functional connectivity from the EEG of drivers in a simulation experiment for driver fatigue classification. In general, the paper is interesting and well-written. The following comments are suggested to consider before it is accepted.

1. Why did the authors divide the specific five frequency bands for removing EEG artifacts?

2. It is suggested to briefly review the feature selection using the spectral estimation method. The authors should explain why they adopted Welch’s method. The adaptive multi-taper method is recommended to pay attention to, and the associated reference ‘An adaptive multi-taper spectral estimation for stationary processes’ is suggested added.

3. How to determine the hyperparameters of the SVM? The hyperparameters directly affect the SVM performance, and the random search, grid search, and Bayesian optimization are commonly used methods, which can be briefly introduced. More details can refer to the Probabilistic framework with Bayesian optimization for predicting typhoon-induced dynamic responses of a long-span bridge. This reference should be also added to illustrate these methods.

Author Response

Please refer to the attached file for the response to Reviewer 3.

Round 2

Reviewer 3 Report

The authors have addressed my previous comments.